# Feature Intertwiner for Object Detection

**Hongyang Li[1], Bo Dai[2], Shaoshuai Shi[1], Wanli Ouyang[3] & Xiaogang Wang[1]**

[1]Department of Electronic Engineering
The Chinese University of Hong Kong
{yangli,ssshi,xgwang}@ee.cuhk.edu.hk

[2]Department of Information Engineering
The Chinese University of Hong Kong
bdai@ie.cuhk.edu.hk

[3]The University of Sydney, SenseTime Computer Vision Research Group
wanli.ouyang@sydney.edu.au

## Abstract

A well-trained model should classify objects with a unanimous score for every category. This requires the high-level semantic features should be alike among samples, despite a wide span in resolution, texture, deformation, *etc*. Previous works focus on re-designing the loss function or proposing new regularization constraints on the loss. In this paper, we address this problem via a new perspective. For each category, it is assumed that there are two sets in the feature space: one with more reliable information and the other with a less reliable source. We argue that the reliable set could guide the feature learning of the less reliable set during training - in the spirit of student mimicking teacher's behavior and thus pushing towards a more compact class centroid in the high-dimensional space. Such a scheme also benefits the reliable set since samples become closer within the same category - implying that it is easier for the classifier to identify. We refer to this mutual learning process as *feature intertwiner* and embed the spirit into object detection. It is well-known that objects of low resolution are more difficult to detect due to the loss of detailed information during network forward pass. We thus regard objects of high resolution as the reliable set and objects of low resolution as the less reliable set. Specifically, an intertwiner is achieved by minimizing the distribution divergence between two sets. We design a historical buffer to represent all previous samples in the reliable set and utilize them to guide the feature learning of the less reliable set. The design of obtaining an effective feature representation for the reliable set is further investigated, where we introduce the optimal transport (OT) algorithm into the framework. Samples in the less reliable set are better aligned with the reliable set with aid of OT metric. Incorporated with such a plug-and-play intertwiner, we achieve an evident improvement over previous state-of-the-arts on the COCO object detection benchmark.

## 1 Introduction

Classifying complex data in the high-dimensional feature space is the core of most machine learning problems, especially with the emergence of deep learning for better feature embedding (Krizhevsky et al., 2012; He et al., 2016; Li et al., 2018) . Previous methods address the feature representation problem by the conventional cross-entropy loss, $l_1$ / $l_2$ loss, or a regularization constraint on the loss term to ensure small intra-class variation and large inter-class distance (Janocha & Czarneck, 2017; Liu et al., 2017b; Wen et al., 2016; Liu et al., 2017a). The goal of these works is to learn more compact representation for each class in the feature space. In this paper, we also aim for such a goal and propose a new perspective to address the problem.

Our observation is that samples can be grouped into two sets in the feature space. One set is more reliable, while the other is less reliable. For example, visual samples may be less reliable due to low resolution, occlusion, adverse lighting, noise, blur, *etc*. The learned features for samples from the reliable set are easier to classify than those from the less reliable one. Our hypothesis is that the

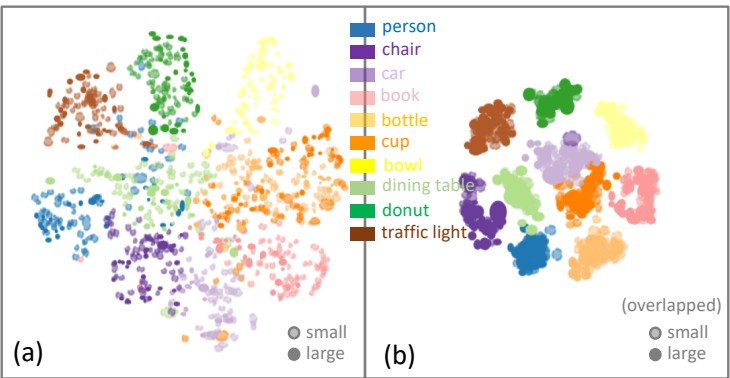

Figure 1: (Zoom in for better view) Visualization of the features in object detection using t-SNE (van der Maaten & Hinton, 2008) (a) without and (b) with feature intertwiner on COCO. Each point is a sample mapped onto the low-dim manifold.

reliable set can guide the feature learning of the less reliable set, in the spirit of a teacher supervising the student. We refer to this mutual learning process as a feature intertwiner.

In this paper, a plug-and-play module, namely, feature intertwiner, is applied for object detection, which is the task of classifying and localizing objects in the wild. An object of lower resolution will inevitably lose detailed information during the forward pass in the network. Therefore, it is well-known that the detection accuracy drops significantly as resolutions of objects decrease. We can treat samples with high resolution (often corresponds to large objects or region proposals) as the reliable set and samples with low resolution (small instances) as the less reliable set[1]. Equipped with these two 'prototypical' sets, we can apply the feature intertwiner where the reliable set is leveraged to help the feature learning of the less reliable set.

Fig. 1 on the left visualizes the learned detection features before classifier[2]. Without intertwiner in (a), samples are more scattered and separated from each other. Note there are several samples that are far from its own class and close to the samples in other categories (*e.g.*, class person in blue), indicating a potential mistake in classification. With the aid of feature intertwiner in (b), there is barely outlier sample outside each cluster. the features in the lower resolution set approach closer to the features in the higher resolution set - achieving the goal of compact centroids in the feature space. Empirically, these two settings correspond to the baseline and intertwiner experiments (marked in gray) in Table 2a. The overall mAP metric increases from 32.8% to 35.2%, with an evident improvement of 2.6% for small instances and a satisfying increase of 0.8% for large counterparts. This suggests the proposed feature intertwiner could benefit both sets.

Two important modifications are incorporated based on the preliminary intertwiner framework. The first is the use of class-dependent historical representative stored in a buffer. Since there might be no large sample for the same category in one mini-batch during training, the record of all previous features of a given category for large instances is recorded by a representative, of which value gets updated dynamically as training evolves. The second is an inclusion of the optimal transport (OT) divergence as a deluxe regularization in the feature intertwiner. OT metric maps the comparison of two distributions on high-dimensional feature space onto a lower dimension space so that it is more sensible to measure the similarity between two distributions. For the feature intertwiner, OT is capable of enforcing the less reliable set to be better aligned with the reliable set.

We name the detection system equipped with the feature intertwiner as **InterNet**. Full code suite is available at https://github.com/hli2020/feature_intertwiner. For brevity, we put the descriptions of dividing two sets in the detection task, related work (partial), background knowledge on OT theory and additional experiments in the appendix.

---

[1]We use the term 'large object/(more) reliable/high resolution set' interchangeably in the following to refer to the same meaning; likewise for the term 'small set/less reliable set/low-resolution set'.

[2]Only top ten categories with the most number of instances in prediction is visualized. For each category, the high-resolution objects (reliable set) are shown in solid color while the low-resolution instances (less reliable set) are shown in transparent color with dashed boundary.

## 2 RELATED WORK

**Object detection** (Dai et al., 2016; He et al., 2017; Girshick, 2015; Ren et al., 2015; Lin et al., 2017a; Redmon & Farhadi, 2016) is one of the most fundamental computer vision tasks and serves as a precursor step for other high-level problems. It is challenging due to the complexity of features in high-dimensional space (Krizhevsky et al., 2012), the large intra-class variation and inter-class similarity across categories in benchmarks (Deng et al., 2009; Tsung-Yi Lin, 2015). Thanks to the development of deep networks structure (Simonyan & Zisserman, 2015; He et al., 2016) and modern GPU hardware acceleration, this community has witnessed a great bloom in both performance and efficiency. **The detection of small objects** is addressed in concurrent literature mainly through two manners. The first is by looking at the surrounding context (Li et al., 2016; Mottaghi et al., 2014) since a larger receptive filed in the surrounding region could well compensate for the information loss on a tiny instance during down-sampling in the network. The second is to adopt a multi-scale strategy (Li et al.; Lin et al., 2017a; Liu et al., 2015; Shrivastava et al., 2016) to handle the scale problem. This is probably the most effective manner to identify objects in various sizes and can be seen in (almost) all detectors. Such a practice is a "sliding-window" version of warping features across different stages in the network, aiming for normalizing the sizes of features for objects of different resolutions. The proposed feature intertwiner is perpendicular to these two solutions. We provide a new perspective of addressing the detection of small objects - leveraging the feature guidance from high-resolution reliable samples.

**Designing loss functions for learning better features.** The standard cross-entropy loss does not have the constraint on narrowing down the intra-class variation. Several works thereafter have focused on adding new constraints to the intra-class regularization. Liu *et al.* (Liu et al., 2017a) proposed the angular softmax loss to learn angularly discriminative features. The new loss is expected to have smaller maximal intra-class distance than minimal inter-class distance. The center loss (Wen et al., 2016) approach specifically learns a centroid for each class and penalizes the distances between samples within the category and the center. Our feature intertwiner shares some spirit with this work in that, the proposed buffer is also in charge of collecting feature representatives for each class. A simple modification (Liu et al., 2017b) to the inner product between the normalized feature input and the class centroid for the softmax loss also decreases the inner-class variation and improves the classification accuracy. Our work is from a new perspective in using the reliable set for guiding the less reliable set.

## 3 FEATURE INTERTWINERS FOR OBJECT DETECTION

In this paper, we adopt the Faster RCNN pipeline for object detection (He et al., 2016; 2017; Girshick, 2015). In Faster RCNN, the input image is first fed into a backbone network to extract features; a region proposal network (Ren et al., 2015) is built on top of it to generate potential region proposals, which are several candidate rectangular boxes that might contain objects. These region proposals vary in size. Then the features inside the region are extracted and warped into the same spatial size (by RoI-pooling). Finally, the warped features are used by the subsequent CNN layers for classifying whether an object exists in the region.

### 3.1 FEATURE INTERTWINER OVERVIEW

We now explicitly depict how the idea of feature intertwiner could be adapted into the object detection framework. Fig. 2 describes the overall pipeline of the proposed InterNet.

A network is divided into several levels based on the spatial size of feature maps. For each level $l$, we split the set of region proposals into two categories: one is the large-region set whose size is larger than the output size of RoI-pooling layer and another the small-region set whose size is smaller. These two sets corresponds to the reliable and less reliable sets, respectively. For details on the generation of these two sets in object detection, refer to Sec. 6.2 in the appendix. Feature map $P_l$ at level $l$ is fed into the RoI layer and then passed onto a *make-up* layer. This layer is designed to fuel back the lost information during RoI and compensate necessary details for instances of small resolution. The refined high-level semantics after this layer is robust to factors (such as pose, lighting, appearance, *etc.*) despite sample variations. It consists of one convolutional layer without

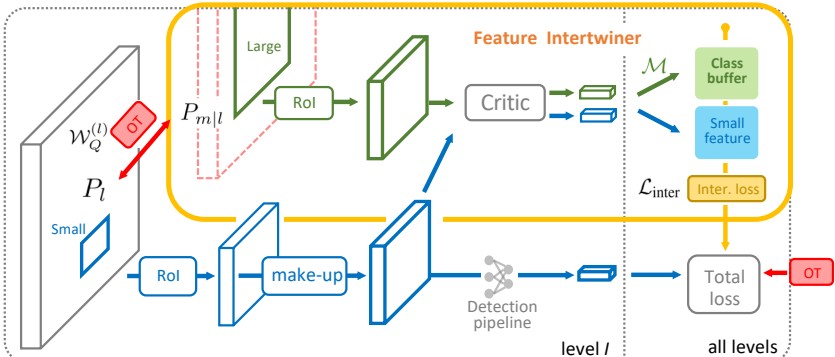

Figure 2: Feature intertwiner overview. Blue blobs stands for the less reliable set (small objects) and green for the reliable set (large ones). For current level $l$, feature map $P_l$ of the small set is first passed into a RoI-pooling layer. Then it is fed into a make-up layer, which fuels back the information lost during RoI; it is optimized via the intertwiner module (yellow rectangle), with aid of the reliable set (green). 'OT' (in red) stands for the optimal transport divergence, which aligns information between levels (for details see Sec. 3.3). $P_{m|l}$ is the input feature map of the reliable set for the RoI layer; $m$ indicates higher level(s) than $l$.

altering the spatial size. The make-up unit is learned and optimized via the intertwiner unit, with aid of features from the large object set, which is shown in the upstream (green) of Fig. 2.

The feature intertwiner is essentially a data distribution measurement to evaluate divergence between two sets. For the reliable set, the input is directly the outcome of the RoI layer of the large-object feature maps $P_{m|l}$, which correspond to samples of higher level/resolution. For the less reliable set, the input is the output of the make-up layer. Both inputs are fed into a *critic* module to extract further representation of these two sets and provide evidence for intertwiner. The critic consists of two convolutions that transfer features to a larger channel size and reduce spatial size to one, leaving out of consideration the spatial information. A simple $l_2$ loss can be used for comparing difference between two sets. The final loss is a combination of the standard detection losses (Girshick, 2015) and the intertwiner loss across all levels.

The detailed network structure of the make-up and critic module in the feature intertwiner is shown in the appendix (Sec. 6.6). There are two problems when applying the aforementioned pipeline into application. The first is that the two sets for the same category often do not occur simultaneously in one mini-batch; the second is how to choose the input source for the reliable set, *i.e.*, feature map $P_{m|l}$ for the large object set. We address these two points in the following sections.

### 3.2 CLASS BUFFER

The goal of the feature intertwiner is to have samples from less reliable set close to the samples *within the same category* from the reliable set. In one mini-batch, however, it often happens that samples from the less reliable set are existent while samples of the same category from the reliable set are non-existent (or vice versa). This makes it difficult to calculate the intertwiner loss between two sets. To address this problem, we use a buffer $\mathcal{B}$ to store the *representative* (prototype) for each category. Basically the representative is the mean feature representation from large instances.

Let the set of features from the large-region object on all levels be $\mathbf{F}_{\text{critic}}^{(\texttt{large})}$; each sample consisting of the large set $\mathbf{F}$ be $\boldsymbol{f}^{(j)}$, where $j$ is the sample index and its feature dimension is $d$. The buffer could be generated as a mapping from sample features to class representative:

$$\mathcal{B} = [\boldsymbol{b}_1, \ldots, \boldsymbol{b}_i, \ldots, \boldsymbol{b}_{N_{\text{cls}}}] = \mathcal{M}\left[\mathbf{F}_{\text{critic}}^{(\texttt{large},1)}, \ldots, \mathbf{F}_{\text{critic}}^{(\texttt{large},l)}, \ldots, \mathbf{F}_{\text{critic}}^{(\texttt{large},L)}\right], \quad (1)$$

$$\boldsymbol{b}_{i^*} = \mathcal{M}\left[\mathbf{F}_{\text{critic}}^{(\texttt{large},l)}\right] = \frac{1}{Z}\sum_{l,j}\boldsymbol{f}_{\text{critic}}^{(\texttt{large},l,j)}, \quad \text{where } \mathbf{F}_{\text{critic}}^{(\texttt{large},l)} = \{\boldsymbol{f}_{\text{critic}}^{(\texttt{large},l,j)} \in \mathbb{R}^d\}, \quad (2)$$

where the total number of classes is denoted as $N_{\text{cls}}$. Each entry $\boldsymbol{b}_i$ in the buffer $\mathcal{B}$ is referred to as the representative of class $i$. Every sample, indexed by $j$ in the large object set, contributes to the class representative $i^*$ if its label belongs to $i^*$. Here we denote $i^*$ as the label of sample $j$; and $Z$ in Eqn. (2) denotes the total number of instances whose label is $i^*$. The representative is deemed as a reliable source of feature representation and could be used to guide the learning of the less reliable set. There are many options to design the mapping $\mathcal{M}$, *e.g.*, the weighted average of all features in the past iterations during training within the class as shown in Eqn. (2), feature statistics from only a period of past iterations, *etc.* We empirically discuss different options in Table 2d.

Equipped with the class buffer, we define the intertwiner loss between two sets as:

$$\mathcal{L}_{\text{inter}} = \sum_{l,j} \mathcal{D}\big(\boldsymbol{f}_{\text{critic}}^{(\texttt{small},l,j)}, \mathcal{B}\big), \tag{3}$$

where $\mathcal{D}$ is a divergence measurement; $\boldsymbol{f}_{\text{critic}}^{(\texttt{small},l,j)}$ denotes the semantic feature after critic of the $j$-th sample at level $l$ in the less reliable set (small instances). Note that the feature intertwiner is proposed to optimize the feature learning of the less reliable set for each level. During inference, the green flow as shown in Fig. 2 for obtaining the class buffer will be removed.

**Discussion on the intertwiner. (a)** Through such a mutual learning, features for small-region objects gradually encode the affluent details from large-region counterparts, ensuring that the semantic features within one category should be as much similar as possible despite the visual appearance variation caused by resolution change. The resolution imperfection of small instances inherited from the RoI interpolation is compensated by mimicking a more reliable set. Such a mechanism could be seen as a teacher-student guidance in the self-supervised domain (Chen et al., 2017). **(b)** It is observed that if the representative $\boldsymbol{b}_i$ is detached in back-propagation process (*i.e.*, no backward gradient update in buffer), performance gets better. The buffer is used as the guidance for less reliable samples. As contents in buffer are changing as training evolves, excluding the buffer from network update would favorably stabilize the model to converge. Such a practice shares similar spirit of the replay memory update in deep reinforcement learning. **(c)** The buffer statistics come from all levels. Note that the concept of "large" and "small" is a *relative* term: large proposals on current level could be deemed as "small" ones on the next level. However, the level-agnostic buffer would always receive semantic features for (strictly) large instances. This is why there are improvements across *all* levels (large or small objects) in the experiments.

### 3.3 Choosing Best Feature Map for Large Objects using Optimal Transport

How to acquire the input source, denoted as $P^{(\texttt{large},l)}$, *i.e.*, feature maps of large proposals, to be fed into the RoI layer on current level $l$? The feature maps, denoted by $P_l$ or $P_m$, are the output of ResNet at different stages, corresponding to different resolutions. Altogether we use four stages, *i.e.*, $P_2$ to $P_5$; $P_2$ corresponds to feature maps of the highest resolution and $P_5$ has the lowest resolution. The inputs are crucial since they serve as the guidance targets to be learned by small instances. There are several choices, which is depicted in Fig. 3.

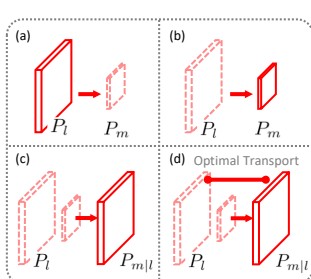

Figure 3: Different designs for the input source in the reliable set. Solid shape is the chosen plan in each option.

| option | variant | AP | AP$_{50}$ | AP$_{75}$ | AP$_S$ | AP$_M$ | AP$_L$ |
|---|---|---|---|---|---|---|---|
| (a) use $P_l$ | | 35.1 | 54.9 | 40.7 | 20.2 | 38.3 | 48.5 |
| (b) use $P_m$ (baseline) | | 40.5 | 62.8 | 47.6 | 23.7 | 45.2 | 53.1 |
| (c) $P_{m\|l}$ | $\mathcal{F}$ bilinear | 40.6 | 62.9 | 47.6 | 23.9 | 45.4 | 53.1 |
| | $\mathcal{F}$ neural net* | 41.3 | 63.5 | 48.5 | 24.6 | 46.3 | 53.8 |
| | *increase from* (b) *to* (c)* | *+0.8* | *+0.7* | *+0.9* | *+0.9* | *+1.1* | *+0.7* |
| (d) $P_{m\|l}$ | KL, $\mathcal{F}$ neural net | 41.0 | 63.1 | 48.2 | 24.5 | 45.7 | 53.4 |
| | $l_2$, $\mathcal{F}$ neural net | 41.8 | 64.2 | 48.9 | 24.7 | 46.0 | 53.8 |
| | optimal transport (OT)** | **42.5** | **65.1** | **49.4** | **25.4** | **46.6** | **54.3** |
| | biased version of OT | 42.5 | 65.3 | 48.6 | 25.3 | 46.8 | 54.3 |
| | *increase from* (b) *to* (d)** | *+2.0* | *+2.3* | *+1.8* | *+1.7* | *+1.4* | *+1.2* |

Table 1: Numeric results on different input sources for the reliable set (using ResNet-101-FPN model). $\mathcal{F}$ is the up-sampling layer; we use option (d), OT as the final candidate. The biased version of optimal transport is detailed in appendix, Sec. 6.3.

**Option (a):** $P^{(\texttt{large},l)} = P_l$**.** The most straightforward manner would be using features on current level as input for large object set. This is inappropriate since $P_l$ is trained in RPN specifically for identifying small objects; adopting it as the source could contain noisy details of small instances.

**Option (b):** $P^{(\texttt{large},l)} = P_m$**.** Here $m$ and $l$ denote the index of stage/level in ResNet and $m > l$. One can utilize the higher level feature map(s), which has the proper resolution for large objects. Compared with $P_l$, $P_m$ have lower resolution and higher semantics. For example, consider the large instances assigned to level $l = 2$ (how to assign large and small instances is discussed in the appendix Sec. 6.2), $P_m$ indicates three stages $m = 3, 4, 5$. However, among these large instances, some of them are deemed as small objects on higher level $m$ - implying that those feature maps $P_m$ might not carry enough information. They would *also* have to be up-sampled during the RoI operation for updating the buffer on current level $l$. Take Table 3 in the appendix for example, among the assigned 98 proposals on level 2, there are 31 (11 on level 3 and 20 on level 4) objects that have insufficient size (smaller than RoI's output). Hence it might be inappropriate to directly use the high-level feature map as well.

**Option (c):** $P^{(\texttt{large},l)} = P_{m|l} \triangleq \mathcal{F}(P_m)$**.** $P_m$ is first up-sampled to match the size at $P_l$ and then is RoI-pooled with outcome denoted as $P_{m|l}$. The up-sampling operation aims at optimizing a mapping $\mathcal{F} : P_m \mapsto P_{m|l}$ that can recover the information of large objects on a shallow level. $\mathcal{F}$ could be as simple as a bilinear interpolation or a neural network.

These three options are empirically reported in Table 1. The baseline model in (b) corresponds to the default setting in cases 2d, 2e of Table 2, where the feature intertwiner is adopted already. There is a 0.8% AP boost from option (b) to (c), suggesting that $P_m$ for large objects should be converted back to the feature space of $P_l$. The gain from (a) to (c) is more evident, which verifies that it might not be good to use $P_l$ directly. More analysis is provided in the appendix.

Option (c) is a better choice for using the reliable feature set of large-region objects. Furthermore, we build on top of this choice and introduce a better alternative to build the connection between $P_l$ and $P_{m|l}$, since the intertwiner is designed to guide the feature learning of the less reliable set on the current level. If some constraint is introduced to keep information better aligned between two sets, the modified input source $P_{m|l}$ for large instance would be more proper for the other set to learn.

**Option (d):** $P^{(\texttt{large},l)} = \texttt{OT}(P_l, P_{m|l})$**.** The spirit of moving one distribution into another distribution optimally in the most effective manner fits well into the optimal transport (OT) domain (Peyr & Cuturi, 2018). In this work, we incorporate the OT unit between feature map $P_l$ and $P_{m|l}$, which serve as inputs before the RoI-pooling operation. A discretized version (Genevay et al., 2017; Cuturi, 2013) of the OT divergence is employed as an additional regularization to the loss:

$$\texttt{OT}(P_l, P_{m|l}) \triangleq \mathcal{W}_Q(\mathbb{P}_\psi, \mathbb{P}_r) \xleftarrow{\text{discrete}} \min_{P \in \mathbb{R}_+^{C_2 \times C_1}} \langle Q, P \rangle, \qquad (4)$$

where the non-positive $P$ serves as a proxy for the coupling and satisfies $P^\top \mathbb{1}_{C_2} = \mathbb{1}_{C_1}, P\mathbb{1}_{C_1} = \mathbb{1}_{C_2}$. $\langle \cdot, \cdot \rangle$ indicates the Frobenius dot-product for two matrices and $\mathbb{1}_m := (1/m, \ldots, 1/m) \in \mathbb{R}_+^m$. Now the problem boils down to computing $P$ given some ground cost $Q$. We adopt the Sinkhorn algorithm (Sinkhorn, 1964) in an iterative manner to compute $\mathcal{W}_Q$, which is promised to have a differentiable loss function. The OT divergence is hence referred to as Sinkhorn divergence.

Given features maps $P_m$ from higher level, the generator network $\mathcal{F}$ up-samples them to match the size of $P_l$ and outputs $P_{m|l}$. The channel dimension of $P_l$ and $P_{m|l}$ is denoted as $C$. The critic unit $\mathcal{H}$ (not the proposed critic unit in the feature intertwiner) is designed to reduce the spatial dimensionality of input to a lower dimension $k$ while keeping the channel dimension unchanged. The number of samples in each distribution is $C$. The outcome of the critic unit in OT module is denoted as $\boldsymbol{p}_l, \boldsymbol{p}_{m|l}$, respectively. We choose cosine distance as the measurement to calculate the distance between manifolds. The output is known as the ground cost $Q_{x,y}$, where $x, y$ indexes the sample in these two distributions. The complete workflow to compute the Sinkhorn divergence is summarized in Alg. 1. Note that each level owns their own OT module $\mathcal{W}_Q^l(P_l, P_m) = \texttt{OT}(P_l, P_{m|l})$. The total loss for the detector is summarized as:

$$\mathcal{L} = \mathcal{L}_{\text{inter}} + \sum_l \mathcal{W}_Q^{(l)}(P_l, P_m) + \mathcal{L}_{\text{standard}}, \qquad (5)$$

where $\mathcal{L}_{\text{standard}}$ is the classification and regression losses defined in most detectors (Girshick, 2015).

---

**Algorithm 1** Sinkhorn divergence $\mathcal{W}_Q$ adapted for object detection (red rectangle in Fig.2)

---

**Input:**  Feature maps on current and higher levels, $P_l, P_m$
     The generator network $\mathcal{F}$ and the critic unit in OT module $\mathcal{H}$
**Output:** Sinkhorn loss $\mathcal{W}_Q^l(P_l, P_m) = \mathtt{OT}(P_l, P_{m|l})$

Upsample via generator $P_{m|l} = \mathcal{F}(P_m)$
Feed both inputs into critic $\boldsymbol{p}_l = \mathcal{H}(P_l), \boldsymbol{p}_{m|l} = \mathcal{H}(P_{m|l})$     $\triangleright \boldsymbol{p}_{(\cdot)}$ size $C \times k$
$\forall (x,y)$ in $\boldsymbol{p}_l, \boldsymbol{p}_{m|l}$, define the ground cost $Q_{x,y} = \mathtt{cosine\_dist}(\boldsymbol{p}_l, \boldsymbol{p}_{m|l})$   $\triangleright Q$ size $C \times C$

Initialize coefficients $b^{(0)} = \mathbb{1}_C$
Compute Gibbs kernel $K_{x,y} = \exp(-Q_{x,y}/\varepsilon)$       $\triangleright$ controlling factor $\varepsilon = 0.1$
**for** $l = 0$ to $L$ **do**               $\triangleright$ iteration budget $L = 10$
  $a^{(l+1)} := \frac{\mathbb{1}_C}{K^\top b^{(l)}}, b^{(l+1)} := \frac{\mathbb{1}_C}{K a^{(l)}},$     $\triangleright$ known as $\mathtt{Sinkhorn\ iterate}$
**end for**
Compute the proxy matrix $P^{(L)} = \mathrm{diag}(b^{(L)}) \cdot K \cdot \mathrm{diag}(a^{(L)})$
Compute $\mathcal{W}_Q$ based on the dot-product in Eqn. (4): $\langle Q, P \rangle$.

---

**Why prefer OT to other alternatives.** As proved in (Arjovsky et al., 2017), the OT metric converges while other variants (KL or JS divergence) do not in some scenarios. OT provides sensible cost functions when learning distributions supported by low-dim manifolds (in our case, $\boldsymbol{p}_l$ and $\boldsymbol{p}_{m|l}$) while other alternatives do not. As verified via experiments in Table 1, such a property could facilitate the training towards a larger gap between positive and false samples. In essence, OT metric maps the comparison of two distributions on high-dimensional feature space onto a lower dimension space. The use of Euclidean distance could improve AP by around 0.5% (see Table 1, (d) $l_2$ case), but does not gain as much as OT does. This is probably due to the complexity of feature representations in high-dimension space, especially learned by deep models.

## 4   EXPERIMENTAL RESULTS

We evaluate InterNet on the object detection track of the challenging COCO benchmark (Tsung-Yi Lin, 2015). For training, we follow common practice as in (Ren et al., 2015; He et al., 2017) and use the `trainval35k` split (union of 80k images from `train` and a random 35k subset of images from 40k `val` split) for training. The lesion and sensitivity studies are reported by evaluating on the `minival` split (the remaining 5k images from `val`). For all experiments, we use depth 50 or 101 ResNet (He et al., 2016) with FPN (Lin et al., 2017a) constructed on top. We base the framework on Mask-RCNN (He et al., 2017) *without* the segmentation branch. All ablative analysis adopt austere settings: training and test image scale only at 512; no multi-scale and data augmentation (except for horizontal flip). Details on the training and test procedure are provided in the appendix (Sec. 6.5).

### 4.1   ABLATION STUDY ON INTERTWINER MODULE

**Baseline comparison.** Table 2a lists the comparison of InterNet to baseline, where both methods shares the same setting. On average it improves by 2 points in terms of mAP. The gain for small objects is much more evident. Note that our method also enhances the detection of large objects (by 0.8%), since the last level also participates in the intertwiner update by comparing its similarity feature to the history buffer, which requires features of the same category to be closer to each other. The last level does not contribute to the buffer update though.

**Assignment strategy** (analysis based on Sec. 6.2). Table 2a also investigates the effect of different region proposal allocations. 'by RoI size' divides proposals whose area is below the RoI threshold in Table 3 as small and above as large; 'more on higher' indicates the base value in Eqn. (6) is smaller (=40); the default setting follows (Lin et al., 2017a) where the base is set to 224. Preliminary, we think putting more proposals on higher levels (the first two cases) would balance the workload of the intertwiner; since the default setting leans towards too many proposals on level 2. However, there is no gain due to the mis-alignment with RPN training. The distribution of anchor templates in RPN does not alter accordingly, resulting in the inappropriate use of backbone feature maps.

| | proposal split | AP | $AP_{50}$ | $AP_{75}$ | $AP_S$ | $AP_M$ | $AP_L$ |
|---|---|---|---|---|---|---|---|
| | by RoI size | 30.9 | 53.7 | 35.1 | 10.8 | 34.7 | 46.6 |
| baseline | more on higher | 31.3 | 54.0 | 35.8 | 11.4 | 35.1 | 47.5 |
| | default * | 32.8 | 55.3 | 37.2 | 12.7 | 36.8 | 49.3 |
| | by RoI size | 33.7 | 56.1 | 37.6 | 13.5 | 37.4 | 50.8 |
| intertwiner | more on higher | 32.3 | 55.7 | 37.1 | 12.9 | 36.2 | 49.5 |
| | default ** | **35.2** | **57.6** | **38.0** | **15.3** | **38.7** | **51.1** |
| *increase from * to** * | | *+2.4* | *+2.1* | *+0.8* | *+2.6* | *+1.9* | *+0.8* |

| | AP | $AP_{50}$ | $AP_{75}$ |
|---|---|---|---|
| $l_1$ | 34.2 | 57.1 | 37.2 |
| $l_2$ (default) | 35.2 | **57.6** | **38.0** |
| KL `div` | 34.6 | 57.8 | 37.4 |
| $l_1$ (fac=10) | 34.4 | 57.6 | 37.8 |
| $l_2$ (fac=0.1) | 34.8 | 58.0 | 37.5 |
| KL `div` (fac=10) | **35.6** | 58.2 | 38.01 |

(a) **Baseline and proposal assignment strategy**: intertwiner increases detection of both small and large objects compared to baseline. Putting more proposals on lower level brings more gain.

(b) **Feature intertwiner loss**: upper block uses a factor of 1.0. $l_2$ performs slightly better than KL divergence.

| | AP | $AP_{50}$ | $AP_{75}$ |
|---|---|---|---|
| separate | 34.0 | 57.1 | 37.3 |
| naive add | | — fail — | |
| linear | **35.2** | **57.6** | **38.0** |

| | size/weight | AP | $AP_{50}$ | $AP_{75}$ |
|---|---|---|---|---|
| partial | 2000 | 37.3 | 58.5 | 44.7 |
| | 15k (epoch) | 38.8 | 59.9 | 46.1 |
| all history | decay weight | 39.2 | 60.6 | 45.4 |
| | equal weight | **40.5** | **62.8** | **47.6** |

| | yes? | AP | $AP_{50}$ | $AP_{75}$ |
|---|---|---|---|---|
| multiple $\mathcal{B}$ | ✓ | 40.58 | 62.83 | 47.62 |
| | ✗ | 40.54 | 62.81 | 47.61 |
| detach $b_i$ | ✓ | 40.5 | 62.8 | 47.6 |
| | ✗ | 40.1 | 62.4 | 47.3 |

(c) **Boosted detection feature source**: merging $f_{\text{critic}}$ into the detection folllowup pipeline increases result.

(d) **Buffer choice design** (101-layer): buffer taking in all history with equal weight ensures best accuracy. Longer size in 'partial' block enhances result and yet possesses more parameters.

(e) **Workflow design** (101-layer): applying different buffers on each level barely matters; detaching $b_i$ during back-propagation is better.

Table 2: Ablation study on the component design of feature intertwiner. Gray background is the final default setting adopted in each case. Network is either ResNet-50-FPN or 101.

**Intertwiner loss.** Upper block in Table 2b shows a factor of 1.0 to be merged on the total loss whereas lower block depicts a specific factor that achieves better AP than others. The simple $l_2$ loss achieves slightly better than the KL divergence, where the latter is computed as $L_{\text{inter}} = b \cdot \log(b/\boldsymbol{f})$. The $l_1$ option is by around 1 point inferior than these two and yet still verifies the effectiveness of the intertwiner module compared with baseline (34.2 *vs* 32.8) - implying the generalization ability of our method in different loss options.

**How does the intertwiner module affect learning?** By measuring the divergence between two sets (*i.e.*, small proposals in the batch and large references in the buffer), we have gradients, as the influence, back-propagated from the critic to make-up layer. In the end, the make-up layer is optimized to enforce raw RoI outputs recovering details even after the loss from reduced resolution. The naive design denoted by 'separate' achieves 34.0% AP as shown in Table 2c. To further make the influence of the intertwiner stronger, we linearly combine the features after critic with the original detection feature (with equal weights, *aka* 0.5; *not* shown in Fig. 2) and feed this new combination into the final detection heads. This improves AP by 1 point (denoted as 'linear' in Table 2c). The 'naive add' case with equal weights 1 does not work (loss suddenly explodes during training), since the amplitude of features among these two sources vary differently if we simply add them.

**Does buffer size matter?** Table 2d shows that it does not. A natural thought could be having a window size of $K$ and sliding the window to keep the most recent features recorded. In general, larger size improves performance (see case '2000' *vs* the size of 'one epoch' where batch size is 8, 37.3% $\rightarrow$ 38.8%). In these cases, statistics of large object features for one category cannot reflect the whole training set and it keeps alternating as network is updated. Using 'all history' data by running averaging not only saves memory but also has the whole picture of the data. Preliminary, we choose a decayed scheme that weighs more to recent features than ones in the long run, hoping that the model would be optimized better as training evolves. However, experiments does not accord with such an assumption: AP is better where features are equally averaged (*c.f.*, 40.5% and 39.2%) in terms of network evolution.

**Unified or level-based buffer?** Unified. Table 2e upper block reports such a perspective. In early experiments, we only have one unified buffer in order to let objects on the last level also involved in the intertwiner. Besides, the visual features of large objects should be irrelevant of scale variation. This achieves a satisfying AP already. We also try applying different buffers on each level[3]. The performance improvement is slight, although the additional memory cost is minor.

---

[3]In such case, the last level adopts the buffer on level 2 since it contains the most number of large objects.

**Other investigations.** As discussed at the end of Sec. 3.1, detaching buffer transaction from gradient update attracts improvement (40.5% *vs* 40.1% in Table 2e). Moreover, we tried imposing stronger supervision on the similarity feature of large proposals by branching out a cross-entropy loss, for purpose of diversifying the critic outputs among different categories. However, it does not work and this additional loss seems to dominate the training process.

## 4.2 COMPARISON TO STATE-OF-THE-ARTS

**Performance.** We list a comparison of our InterNet with previous state-of-the-arts in Table 4 in the appendix. Without multi-scale technique, ours (42.5%) still favorably outperforms other two-stage detectors (*e.g.*, Mask-RCNN, 39.2%) as well as one-stage detector (SSD, 31.2%). Moreover, we showcase in Fig. 4 the per-class improvement between the baseline and the improved model after adopting feature intertwiner in Table 2a (two gray rows). The most improved classes are microwave, truck while the results in couch, bat decrease. Most small-size categories get improved. As for the

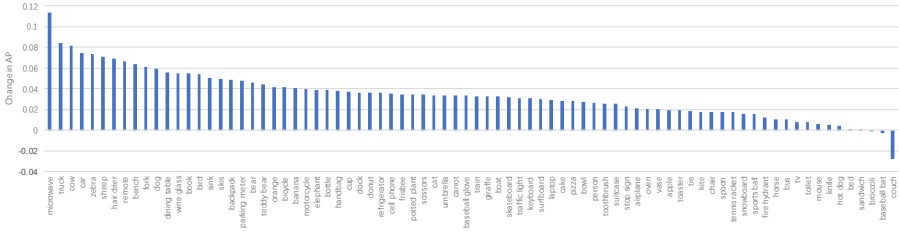

Figure 4: Improvement per category after embedding the feature intertwiner on COCO dataset.

distinct drop for the 'couch' class, we find that for a large couch among samples on COCO, usually there sit a bunch of people, stuff, pets, *etc*. And yet the annotations in these cases would cover the whole scenario including these noises, making the feature representation of the large couch quite inaccurate. The less accurate features would guide the learning of their small counterparts, resulting in a lower AP for this class.

**Model complexity and timing.** The feature intertwiner only increases three light-weight conv. layers at the make-up and critic units. The usage of class buffer could take up a few GPU memory on-the-fly; however, since we adopt an 'all-history' strategy, the window size is just 1 instead of a much larger $K$. The additional cost to the overall model parameters is also from the OT module for each level; however, we find using just one conv. layer for the critic $\mathcal{H}$ and two conv. layers with small kernels for generator $\mathcal{F}$ is enough to achieve good result. Training on 8 GPUs with batch size of 8 takes around 3.4 days; this is slower than Mask-RCNN reported in (He et al., 2017). The memory cost on each card is 9.6 GB, compared with baseline 8.3 GB. The inference runs at 325ms per image (input size is 800) on a Titan Pascal X, increasing around 5% time compared to baseline (308 ms). We do not intentionally optimize the codebase, however.

## 5 CONCLUSION AND FUTURE WORK

In this paper, we propose a feature intertwiner module to leverage the features from a more reliable set to help guide the feature learning of another less reliable set. This is a better solution for generating a more compact centroid representation in the high-dimensional space. It is assumed that the high-level semantic features within the same category should resemble as much as possible among samples with different visual variations. The mutual learning process helps two sets to have closer distance within the cluster in each class. The intertwiner is applied on the object detection task, where a historical buffer is proposed to address the sample missing problem during one mini-batch and the optimal transport (OT) theory is introduced to enforce the similarity among the two sets. Since the features in the reliable set serve as teacher in the feature learning, careful preparation of such features is required so that they would match the information in the small-object set. This is why we design different options for the large set and finally choose OT as a solution. With aid of the feature intertwiner, we improve the detection performance by a large margin compared to previous state-of-the-arts, especially for small instances.

Feature intertwiner is positioned as a general alternative to feature learning. As long as there exists proper division of one reliable set and the other less reliable set, one can apply the idea of utilizing the reliable set guide the feature learning of another, based on the hypothesis that these two sets share similar distribution in some feature space. One direction in the future work would be applying feature intertwiner into other domains, *e.g.*, data classification, if proper set division are available.

ACKNOWLEDGMENTS

We thank Buyu Li for helpful comments in a preliminary version of this work. H. Li and S. Shi are supported by the Hong Kong PhD Fellowship Scheme. This project is also supported by the Research Grants Council of Hong Kong under grant CUHK14208417, CUHK14202217, and the Hong Kong Innovation and Technology Support Programme Grant ITS/121/15FX.

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

## 6 APPENDIX

### 6.1 MORE RELATED WORK

**Self-supervised learning.** The buffer in the feature intertwiner can be seen as utilizing non-visual domain knowledge on a set of data to help supervise the feature learning for another set in high-dimensional space. Such a spirit falls into the self-supervised learning domain. In (Chen et al., 2017), Chen *et al.* proposed a knowledge distillation framework to learn compact and accurate object detectors. A teacher model with more capacity is designed to provide strong information and guide the learning of a lite-weight student model. The center loss (Wen et al., 2016) is formulated to learn a class center and penalize samples that have a larger distance with the centroid. It aims at enlarging inter-class resemblance with cross-entropy (CE) loss as well as narrowing down inner-class divergence for face recognition. In our work, the feature intertwiner gradually aggregates statistics of a meta-subset and utilizes them as targets during the feature learning of a less accurate (yet holding a majority) subset. We are inspired by the proposal-split mechanism in object detection domain to learn recognition at separate scales in the network. The **self-paced learning** framework (Kumar et al., 2010) deals with two sets as well, where the easy examples are first introduced to optimize the hidden variable and later on during training, the hard examples are involved. There is no interaction between the two sets. The division is based on splitting different samples. In our framework, the two sets mutually help and interact with each other. The goal is towards optimizing a more compact class centroid in the feature space. These are two different branches of work.

**Optimal transport (OT)** has been applied in two important tasks. One is for transfer learning in the domain adaption problem. Lu *et al.* (Lu et al., 2017) explored prior knowledge in the cost matrix and applied OT loss as a soft penalty for bridging the gap between target and source predictions. Another is for estimating generative models. In (Salimans et al., 2018), a metric combined with OT in primal form with an energy distance results in a highly discriminative feature representation with unbiased gradients. Genevay *et al.* (Genevay et al., 2017) presents the first tractable method to train large-scale generative models using an OT-based loss. We are inspired by these works in sense that OT metric is favorably competitive to measure the divergence between two distributions supported on low-dimensional manifolds.

### 6.2 ASSIGNMENT OF LARGE AND SMALL SETS IN OBJECT DETECTION

In this paper we adopt the ResNet model (He et al., 2016) with feature pyramid dressings (Lin et al., 2017a) constructed on top. It generates five levels of feature maps to serve as inputs for the subsequent RPN and detection branches. Denote the level index as $l = \{1, \ldots, 5\}$ and the corresponding feature maps as $P_l$. Level $l = 1$ is the most shallow stage with more local details for detecting tiny objects and level $l = 5$ is the deepest stage with high-level semantics.

Let $\mathcal{A} = \{a_j\}$ denote the whole set of proposals generated by RPN from $l_2$ to $l_6$ (level six is generated from $l_5$, for details refer to (Lin et al., 2017a)). The region proposals are divided into different levels from $l_2$ to $l_5$:

$$a_j^{(l)} \to l = a_0 + \log(\sqrt{\text{Area}(a_j)}/\texttt{base}), \tag{6}$$

where $a_0$=4 as in (Lin et al., 2017a); $\texttt{base}$=224 is the canonical ImageNet pre-training setting.

Table 3 shows a detailed breakdown[4] of the proposal allocation based on Eqn. (6). We can see most proposals from RPN focus on identifying small objects and hence are allocated at shallow level $l = 2$. The threshold is set to be the ratio of RoI output's area over the area of feature map. For example, threshold on $l = 3$ is obtained by $(14/64)^2$, where 14 is the RoI output size as default setting. Proposals whose area is *below* the threshold suffer from the inherent design during RoI operation - these feature outputs are up-sampled by a simple interpolation. The information of small regions is already lost and RoI layer does not help much to recover them back. As is shown on the fourth row ("below # / above #"), such a case holds the majority. This observation brings in the necessity of designing a meta-learner to provide guidance on feature learning of small objects due to the loophole during the RoI layer.

---

[4]Each sample has 200 proposals with input size being 512. Batch size is 2, resulting in 400 proposals in total. Statistics are *averaged per iteration*, based on the output of RPN network during training.

| level $l$ | 2 | 3 | 4 | 5 |
|---|---|---|---|---|
| proposal # (perc.) | 302 (75%) | 36 (9%) | 54 (14%) | 8 (2%) |
| threshold | 0.012 | 0.0479 | 0.1914 | 0.7657 |
| below # / above # | 263 / 39 | 25 / 11 | 34 / 20 | 8 / 0 |
| intertwiner small # | 302 | 36 | 54 | 8 |
| intertwiner large # | 98 | 62 | 8 | - |

Table 3: Proposal assignment on each level before RoI operation. 'below #' indicates how many proposals are there whose size is below the size of RoI output. 'intertwiner large #' stands for how many proposals are used for supervising the learning of small objects.

For level $l$ in the network, we define `small` proposals (or RoIs) to be those already assigned by (6) and `large` to be those above $l$:

$$a^{(l,\text{s})} \leftarrow a_j^{(l)}, \quad a^{(l,\text{b})} = \bigcup_{m>l} a_j^{(m)}, \tag{7}$$

where the superscript s,b denotes the set of small and large proposals, respectively. The last two rows in Table 3 show an example of the assignment. These RoIs are then fed into the RoI-pooling layer[5] to generate output features maps for the subsequent detection pipeline to process.

One may wonder the last level do not have large objects for reference based on Eqn. (7). In preliminary experiments, leaving proposals on the last level out of the intertwiner could already improve the overall performance; however, if the last level is also involved (since the buffer is shared across all levels), AP for large objects also improves. See the experiments in Sec. 4.1 for detailed analysis.

## 6.3 SINKHORN DIVERGENCE

Let $u', u$ indicate the individual sample after degenerating high-dimensional features $P_{m|l}, P_l$ from two spaces into low manifolds. $u', u$ are vectors of dimension $k$. The number of samples in these two distributions is denoted by $C_1$ and $C_2$, respectively. The OT metric between two joint probability distributions supported on two spaces $(\mathcal{U}, \mathcal{U})$ is defined as the solution of the linear program (Cuturi, 2013). Denote the data and reference distribution as $\mathbb{P}_\psi, \mathbb{P}_r \in \text{Prob}(\mathcal{U})$[6], respectively, we have the continuous form of OT divergence:

$$\mathcal{W}_Q(\mathbb{P}_\psi, \mathbb{P}_r) = \inf_{\gamma \in \Gamma(\mathbb{P}_\psi, \mathbb{P}_r)} \mathbb{E}\left[ \int_{\mathcal{U} \times \mathcal{U}} Q(u', u) d\gamma(u', u) \right], \tag{8}$$

where $\gamma$ is a coupling; $\Gamma$ is the set of couplings that consists of joint distributions.

Intuitively, $\gamma(u', u)$ implies how much "mass" must be transported from $u'$ to $u$ in order to transform the distribution $\mathbb{P}_\psi$ into $\mathbb{P}_r$; $Q$ is the "ground cost" to move a unit mass. Eqn. (8) above becomes the $p$-Wasserstein distance (or loss, divergence) between probability measures when $\mathcal{U}$ is equipped with a distance $\mathcal{D}_\mathcal{U}$ and $Q = \mathcal{D}_\mathcal{U}(u', u)^p$, for some exponent $p$.

The biased version of Sinkhorn divergence used in Table 1 is defined by:

$$2\mathcal{W}_Q(\mathbb{P}_\psi, \mathbb{P}_r) - \mathcal{W}_Q(\mathbb{P}_r, \mathbb{P}_r) - \mathcal{W}_Q(\mathbb{P}_\psi, \mathbb{P}_\psi).$$

**More analysis on Table 1**. All these options have been discussed explicitly at the beginning of Sec. 3.3. Option (a) is inferior due to the inappropriateness of feature maps; (b) serves as the baseline and used as the default setting in Table 2. Options in (c) verifies that up-sampling feature maps from higher-level onto current level is preferable; $\mathcal{F}$ being a neural net ensures better improvement. Options in (d) illustrates the case where a supervision signal is imposed onto pair $(P_l, P_{m|l})$ to make better alignment between them. We can observe that OT outperforms other variants in this setup. Moreover, we tried a biased version (Genevay et al., 2017) of the Sinkhorn divergence. However, it does not bring in much gain compared to the previous setup. Besides, it could burden system efficiency during training (although it is minor considering the total time per iteration). Such a

---

[5]In this paper, we opt for the RoIAlign (He et al., 2017) option in the RoI layer; one can resort to other options nonetheless. We use term RoI layer, RoI-pooling layer, RoI operation, to refer to the same process.

[6]Prob($\mathcal{U}$) is the set of probability distributions over a metric space $\mathcal{U}$.

phenomenon could result from an improper update of critic and generator inside the OT module, since the gradient flow would be iterated twice more for the last two terms above.

**Extending OT divergence to image classification.** We also testify OT divergence on CIFAR-10 (Krizhevsky & Hinton, 2009) where feature maps between stages are aligned via OT. Test error decreases by around 1.3%. This suggests the potential application of OT in various vision tasks. Different from OT in generative models, we deem the channel dimension as different samples to compare, instead of batch-wise manner as in (Salimans et al., 2018); and treat the optimization of $\mathcal{F}$ and $\mathcal{H}$ in a unified $\min$ problem, as opposed to the adversarial training (Genevay et al., 2017).

## 6.4 COMPARISON TO STATE-OF-THE-ARTS ON COCO AND PASCAL VOC

Table 4 reports the performance of our model compared with other state-of-the-arts on COCO dataset. We can observe that it outperforms all previous one-stage or two-stage detectors by a large margin. The multi-scale technique bundled with data augmentation increases detection accuracy in a more evident manner, which is commonly adopted in most detectors. The updated result in Mask-RCNN is reported as well. It increases the original performance from 38.2% to 43.5% by switching the backbone structure to ResNetX, an updated baseline model, ImageNet-5k pre-training and train-time augmentation. It is better than ours (42.5% without multi-scale version). This is probably mainly due to the change of network structure. Our multi-scale version (44.2%) is better than the updated Mask-RCNN result, however.

| | backbone | AP | $AP_{50}$ | $AP_{75}$ | $AP_S$ | $AP_M$ | $AP_L$ |
|---|---|---|---|---|---|---|---|
| *One-stage detector* | | | | | | | |
| YOLOv2 (Redmon & Farhadi, 2016) | DarkNet-19 | 21.6 | 44.0 | 19.2 | 5.0 | 22.4 | 35.5 |
| SSD513 (Liu et al., 2015) | ResNet-101-SSD | 31.2 | 50.4 | 33.3 | 10.2 | 34.5 | 49.8 |
| DSSD513 (Fu et al., 2017) | ResNet-101-DSSD | 33.2 | 53.3 | 35.2 | 13.0 | 35.4 | 51.1 |
| *Two-stage detector* | | | | | | | |
| F-R-CNN+++ (He et al., 2016) | ResNet-101-C4 | 34.9 | 55.7 | 37.4 | 15.6 | 38.7 | 50.9 |
| F-R-CNN w FPN (Lin et al., 2017a) | ResNet-101-FPN | 36.2 | 59.1 | 39.0 | 18.2 | 39.0 | 48.2 |
| F-R-CNN by G-RMI (Huang et al., 2017) | Incept.-ResNet-v2 | 34.7 | 55.5 | 36.7 | 13.5 | 38.1 | 52.0 |
| F-R-CNN w TDM (Shrivastava et al., 2016) | Incept.-ResNet-v2-TDM | 36.8 | 57.7 | 39.2 | 16.2 | 39.8 | 52.1 |
| R-FCN (Dai et al., 2016) | ResNet-101 | 29.9 | 51.9 | - | 10.8 | 32.8 | 45.0 |
| Mask RCNN (He et al., 2017) | ResNet-101-FPN | 38.2 | 60.3 | 41.7 | 20.1 | 41.1 | 50.2 |
| RetinaNet (Lin et al., 2017b) | ResNet-101-FPN | 39.1 | 59.1 | 42.3 | 21.8 | 42.7 | 50.2 |
| Mask RCNN, updated in (He et al., 2017) | ResNetX-101-FPN | 43.5 | 65.9 | 47.2 | - | - | - |
| **InterNet** (ours) | ResNet-101-FPN | 42.5 | 65.1 | 49.4 | 25.4 | 46.6 | 54.3 |
| **InterNet** (ours) multi-scale | ResNet-101-FPN | **44.2** | **67.5** | **51.1** | **27.2** | **50.3** | **57.7** |

Table 4: Object detection *single-model* performance (bounding box AP) on the COCO `test-dev`. We show two versions of InterNet that incorporates both the feature intertwiner module and OT agreement. The latter is achieved with data augmentation, $1.5\times$ longer training time and multi-scale training. 'F-R-CNN' stands for Faster R-CNN. Our InterNet is also a two-stage detector.

| Model | Structure | Training data | mAP |
|---|---|---|---|
| Fast R-CNN (Girshick, 2015) | VGG-16 | 07 | 66.9 |
| Faster R-CNN (He et al., 2016) | VGG-16 | 07 | 69.9 |
| SSD512 (Liu et al., 2015) | VGG-16 | 07 | 71.6 |
| InterNet (ours) | VGG-16 | 07 | 73.1 |
| Faster R-CNN (He et al., 2016) | ResNet-101 | 07+12 | 76.4 |
| R-FCN (Dai et al., 2016) | ResNet-101 | 07+12 | 80.5 |
| InterNet (ours) | ResNet-101 | 07+12 | 82.7 |

Table 5: Comparison of our model with feature intertwiner to other methods on PASCAL VOC 2007 test set. Here we adopt two backbone options: ResNet-101 and VGG-16 without FPN to fairly compare with others. The number of levels is 4, the same as on COCO benchmark.

To further verify the effectiveness of the feature intertwiner, we further conduct experiments on the PASCAL VOC 2007 dataset. The results are shown in Table 5. Two network structures are adopted.

For ResNet-101, the division of the four levels are similar as ResNet-101-FPN on COCO; for VGG-16, we take the division similarly as stated in SSD (Liu et al., 2015). Specifically, the output of layer 'conv7', 'conv8_2', 'conv9_2' and 'conv10_2' are used for $P_2$ to $P_5$, respectively. Our method performs favorably against others in both backbone structures on the PASCAL dataset.

## 6.5 TRAINING AND TEST DETAILS

We adopt the stochastic gradient descent as optimizer. Initial learning rate is 0.01 with momentum 0.9 and weight decay 0.0001. Altogether there are 13 epoches for most models where the learning rate is dropped by 90% at epoch 6 and 10. We find the warm-up strategy (Goyal et al., 2017) barely improves the performance and hence do not adopt it. The gradient clip is introduced to prevent training loss to explode in the first few iterations, with maximum gradient norm to be 5. Batch size is set to 8 and the system is running on 8 GPUs.

The object detector is based on Mask-RCNN (or Faster-RCNN). RoIAlign is adopted for better performance. The model is initialized with the corresponding ResNet model pretrained on ImageNet. The new proposed feature intertwiner module is trained from scratch with standard initialization. The basic backbone structure for extracting features is based on FPN network (Lin et al., 2017a), where five ResNet blocks are employed with up-sampling layers. The region proposal network consists of one convolutional layer with one classification and regression layer. The classifier structure is similar as RPN's - one convolution plus one additional classification/regression head.

Non-maximum suppression (NMS) is used during RPN generation and detection test phase. Threshold for RPN is set to 0.7 while the value is 0.3 during test. We do not adopt a dense allocation of anchor templates as in some literature (Liu et al., 2015; Redmon et al., 2016); each pixel on a level only has the number of anchors the same as the number of aspect ratios (set to 0.5, 1 and 2). Each level $l$ among the five stages owns a unique anchor size: 32, 64, 128, 256, and 512.

## 6.6 NETWORK STRUCTURE IN FEATURE INTERTWINER

The detailed network architecture on the make-up layer and critic layer are shown below.

| Output size | Layers in the make-up module |
|---|---|
| $B \times C_l \times 14 \times 14$ | conv2d($C_l, C_l, k = 3,$ padding $= 1$) |
| $B \times C_l \times 14 \times 14$ | batchnorm2d($C_l$) |
| $B \times C_l \times 14 \times 14$ | relu($\cdot$) |

Table 6: Network structure of the make-up unit, which consists of one convolutional layer without altering the spatial size. Input: RoI output of the small-set feature map $P_l$. We denote the output of the make-up layer as $P_l'$. $B$ is the batch size in one mini-batch; $C_l$ is the number of channels after the feature extractor in ResNet blocks for each level. For example, when $l = 2, C_l = 256, etc$.

| Output size | Layers in the critic module |
|---|---|
| $B \times 512 \times 7 \times 7$ | conv2d($C_l, 512, k = 3,$ padding $= 1,$ stride $= 2$) |
| $B \times 512 \times 7 \times 7$ | batchnorm2d(512) |
| $B \times 512 \times 7 \times 7$ | relu($\cdot$) |
| $B \times 1024 \times 1 \times 1$ | conv2d(512, 1024, $k = 7$) |
| $B \times 1024 \times 1 \times 1$ | batchnorm1d(1024) |
| $B \times 1024 \times 1 \times 1$ | relu($\cdot$) |
| $B \times 1024 \times 1 \times 1$ | sigmoid($\cdot$) |

Table 7: Network structure of the critic unit. Input: for large set, it is the RoI output of the large-set feature map $P_{m|l}$ and for small set, it is the output of the make-up layer $P_l'$. $B$ is the batch size in one mini-batch; $C_l$ is the number of channels in ResNet blocks.

