# OpenReview forum: "Feature Intertwiner for Object Detection"
_ICLR.cc/2019/Conference_

### Official Review · AnonReviewer1 · 2018-11-02

**Rating:** 7
**Confidence:** 3

**Review:**

This paper proposes a novel approach with the hypothesis that the reliable features can guide the less reliable ones. This approach is applied to the object detection task and show consistent performance improvements.

pros)
(+) This paper is well-written and easy to follow.
(+) The base idea that divides the learned features into two sets; the reliable feature set and the less reliable one is very interesting and looks novel. Plus, the hypothesis, which is that reliable features can guide the features in the less reliable set is also interesting.
(+) The performance improvements are quite large.
(+) Extensive ablative studies are provided to support the proposed method well.

cons)
(-) The method of obtaining the representative in buffer B is not clearly presented.
(-) The overall training and inference procedure are not clearly presented.
(-) Some notations and descriptions are vague and confusing.
(-) More than two datasets are necessary to show the effectiveness of the methods

comments)
- What is the higher level feature map P_m? and How did you choose the higher level feature map at the m-th level in option (b) and (c) in Section 3.3.
- What is the meaning of the "past" features in Section 3.2?
- It is better to show the exact architecture of the make-up module and the critic module.
- Can this method apply to the other backbones such as VGG or ResNets without FPN?
- The sentences at the bottom of p.4 starting with "Note that only~" looks ambiguous.
- f_critic^j may be the j-th element of F_critic, please denote what f_critic^j stands for.

Even if the paper needs to be revised for better readability, I think this paper is above the standard of ICLR because the idea is interesting and novel. Furthermore, the experimental studies are properly designed and well support the main idea. I am leaning toward acceptance, but I would like to see the other reviewers' comments.

---

> ### Author Response · Authors · 2018-11-24
> **Thank you for your feedback (R1)!!!**
>
> Thank you for the positive and helpful comments!!! We really appreciate it and have uploaded a newer version of the manuscript (fonts marked as blue where there is a major change).
>
> >>> The method of obtaining the representative in buffer is not clearly presented; f_critic^j may be the j-th element of F_critic, please denote what f_critic^j stands for.
>
> [RESPONSE]: We have modified the manuscript in Sec. 3.2 to better explain how the representative (b_i) is obtained. It is the average feature representation of one class, taken samples (the critic output) from all the large objects as training evolves. Specifically, we add a new Eqn. (2) besides the original Eqn. (1) to mathematically explain how the representative (b_i) is computed.
>
> f_critic^j is indeed the j-th element (entry) of F_critic:
>
> F_critc = {f_critic^0, f_critic_^1, …, f_critic^j, ...}          (no notation of “large/small” and “l” for brevity here)
>
> Please refer to the new Section 3.2.
>
> >>> Overall training and inference procedure are not clearly presented.
>
> [RESPONSE]: Sorry for leaving reviewer such an impression. We do have a “training and test details” section (Section 6.5) in the appendix. The room for the main paper might be tight so we put it apart from the main paper. We have added more details about the overall network structure in Sec 6.5.
>
> >>> What’s the higher level feature map P_m? How did you choose the higher level feature map at the m-th level in option (b) and (c) in Section 3.3; what is the meaning of the “past” features in Section 3.2?
>
> [RESPONSE]: The higher level feature maps P_m serve as the input source for the RoI layer in the large-object set. “m (or l)” denotes the feature map output in different ResNet stages (corresponding to different resolutions). In ResNet, there are five stages (l or m = 1, ...5); one stage might have a bunch of resnet blocks. The higher one stage is, the higher level semantics it stands for. We use last four levels/stages (P_2, …, P_5) to construct P_m (or P_l). l is the current level and m is the level that is higher than l. For example, if l = 3, then m = 4 and 5.
>
> We have modified Sec. 3.3 to add a footnote about the meaning of P_m and how we choose P_m accordingly in the “Option (b) ….” paragraph (at the end of page 5).
>
> “Past” features in Section 3.2. It means we take the average of features in all large objects that the network has seen in the past mini-batch/iteration training. We revised the sentence a little bit to make it more clear (last few sentences in the paragraph below Enq. (2)).
>
> >>> Better to show the exact structure of the make-up module and critic module.
>
> [RESPONSE]: We have added a separate section (Section 6.6 Network structure in feature intertwiner) in the appendix as requested. And we add a sentence in the main paper (Sec 3.1) to let readers know about this.
>
> >>> The sentences at the bottom of p.4 starting with "Note that only~" looks ambiguous.
>
> [RESPONSE]: This sentence is for making sure the audience understands that the feature intertwiner is not applied during inference, since we don’t know which objects are assigned as large objects for the current level. Thus there is no way to compute the buffer. Only the blue part (for the small object stream) in Figure 2 will be applied during inference. We have shortened the sentence to make it more clear.
>
> >>> Can this method apply to the other backbones such as VGG or ResNets without FPN? More than two datasets are necessary to show the effectiveness of the methods.
>
> [RESPONSE]: Yes, it can be applied to other backbones with FPN, since the assignment of large/small set is independent of the specific structure.
>
> As reviewer requested, we have adopted two additional backbone structures (resnet without FPN and VGG-16) to verify the effectiveness of our feature intertwiner on PASCAL VOC 2007. The results are shown in Table 5 in the appendix. For ResNet-101, the division of the four levels are similar as ResNet-101-FPN on COCO; for VGG-16, we take the division similarly as stated in SSD.  Specifically, the output of layer ‘conv7’, ‘conv8_2’, ‘conv9_2’ and ‘conv10_2’ are used for P_2 to P_5, respectively.
>
> Our method performs favorably against others in both backbone structures on the PASCAL dataset. For example, using VGG-16, we achieve a mAP of 73.1%, compared with SSD 71.6%, Faster-RCNN 69.9% and Fast R-CNN 66.9%.

---

### Official Review · AnonReviewer3 · 2018-11-02
**An interesting take on the problem of detecting small objects**

**Rating:** 9
**Confidence:** 4

**Review:**

OVERVIEW:
The authors tackle the problem of detecting small/low resolution objects in an image. Their key idea is that detecting bigger objects is an easier task and can be used to guide the detection of smaller objects. This is done using the "Feature Intertwiner"  which consists of two branches, one for the larger objects (more reliable set that is also easier to detect) and one for the smaller objects (less reliable set). The second branch contains a make-up layer learned during training (which acts as the guidance from the more reliable set) that helps compensate details needed for detection. The authors define a class buffer that contains representative elements of object features from the reliable set for every category & scale and an intertwiner loss that computes the L2 loss between the features from the less reliable set & the class buffer. They also use an Optimal Transport procedure with a Sinkhorn divergence loss between object features from both sets. The overall loss of the system is now a sum of the detection loss, the intertwiner loss and the optimal transport loss. They evaluate their model on the COCO Object detection challenge showing state-of-the-art performance. They also provide thorough ablation analysis of various design choices. The qualitative result in Fig.1 showing well clustered features for both high & low resolution objects via t-SNE is a nice touch.

COMMENTS:
Clarity - The paper is well written and easy to follow.
Originality & Significance - The paper tackles an important problem and provides a novel solution.
Quality - The paper is complete in that it tackles an important problem, provides a novel solution and demonstrates via thorough experiments the improvement achieved using their approach.

QUESTIONS:
1. The Class Buffer seems very restricted in having a single element per object category per scale to represent all features. The advantage of forcing such a representation is tight clustering in the feature space. But, wouldn't a dictionary approach with multiple elements give more flexibility to the model and learn a richer feature representation at the cost of not-so-good clustering ?
2. Any comment on why you drop performance for couch ? (and baseball bat + bedroll)
3. In Table 4 of Appendix where you compare with more object detection results, I find it interesting that Mask RCNN, updated results has a might higher AP_S (43.5) compared to you (27.2) and everyone else. I was expecting you to be the best under that metric due to the explicit design for small objects. They (MaskRCNN, updated results) are also significantly better than the rest under AP_M but worse under AP_L. Can you explain this behavior ? Is the ResNeXt backbone that much better for small objects ?

---

> ### Author Response · Authors · 2018-11-24
> **Thanks for the comments (R3)!!!**
>
> Thank you for the positive and helpful comments!!! We really appreciate it and have uploaded a newer version of the manuscript (fonts marked as blue where there is a major change).
>
> >>> Class buffer seems very restricted in having a single element per category per scale to represent all features. A dictionary approach with multiple elements give more flexibility to the model and learn a richer representation at the cost of not-so-good clustering?
>
> [RESPONSE]: by saying “a single element” in the class buffer, the reviewer might refer to the output of the critic in Figure 2 (a 1024-dim feature vector for each category) as a single element, if we understand correctly.
>
> One possible solution to implement the reviewer’s suggestion would be, given the critic output - the feature representation for each category, we further feed it into a GMM (Gaussian Mixture Models) to generate different modes (“different elements” in reviewer’s language) for each category, in hope that the GMM outputs would enrich the representation and give more flexibility to the model. Well there are several concerns in this alternative for the buffer design.
>
> (1) First, the number of modes (number of mixtures) for each class is hard to define. For example, a “kite” might be easy to recognize as long as we have sort of “appearance” mode; a “train” category might need more modes than “appearance” since it can be visually very close to “bus”.
>
> (2) Let’s say, the number of modes/elements is set to be the same across all categories, denoted as K. The output of GMM would be K vectors, each with D dimensions. We then compare the difference of distributions between the large and small set. We doubt it would work better since it complicates the case in feature space with K*D-dim features, compared to the current design of D-dim.
>
> >>> Why drop performance for couch? (and baseball bat + bedroll)
>
> [RESPONSE]: Aha, good catch! The “couch” category extremely degrades the performance (~2.x% drop) while the drop for “baseball bat” and “broccoli” is minor. After browsing the images that contain “couch” (http://cocodataset.org/#explore), we find that for a large couch in the image, usually there sit a bunch of people, stuff, pets. And yet the annotations in these cases would cover the whole scenario including these “noises”, making the feature representation of the large couch, quite inaccurate. The less accurate features would guide the learning of their small counterparts, making AP of this class unreliable.
>
> For some examples here:
> http://cocodataset.org/#explore?id=445620
> http://cocodataset.org/#explore?id=346272
> http://cocodataset.org/#explore?id=128113
>
> We have added the discussion above following the first paragraph in Sec. 4.2.
>
>
> >>> Result of “Mask RCNN, updated result” in Appendix, Table 4
>
> [RESPONSE]: It was mistakenly reported in the table. Our bad. The numbers shown in the appendix “39.2 62.5 41.6 || 43.5 65.9 47.2” is the result taken from the fifth row of Table 8 in the Mask-RCNN paper (https://arxiv.org/pdf/1703.06870.pdf). The first three numbers are actually the *segmentation* mAP, which is irrelevant of our task while the last three numbers are the desired *bounding box* mAP. Hence the correct number should be:
>
> Mask RCNN, updated result 	|  ResNetX-101-FPN   |   43.5    65.9   47.2   |  -         -          -
> InterNet (ours)			        |  ResNet-101-FPN     |    42.5    65.1  49.4   | 25.4   46.6    54.3
> InterNet (multi-scale)                 |  ResNet-101-FPN     |    44.2    67.5   51.1  | 27.2   50.3    57.7
>
> Since we don’t find the 43.5% model in the official model zoo (https://github.com/facebookresearch/Detectron/blob/master/MODEL_ZOO.md), the evaluation of the small, medium and large object is thus missing. But based on the mAP of 43.5%, which is better than ours without multi-scale (42.5%), we guess it would be better in some metrics.
>
> The updated Mask-RCNN result increases the original performance from 38.2% to 43.5% by switching the backbone structure to ResNetX, an updated baseline model, ImageNet-5k pre-training and train-time augmentation. It is better than ours and this is probably due to the change of network structure. Our multi-scale version (44.2%) is better than the updated Mask-RCNN result, however.
>
> We have correctified Table 4 in the appendix and added the discussion above in the first paragraph of Sec. 6.4.

---

### Official Review · AnonReviewer2 · 2018-11-04
**performance is good but novelty is limited**

**Rating:** 5
**Confidence:** 4

**Review:**

This paper aims to facilitate feature learning in NN models by exploiting more from reliable examples. This is very similar to self-paced learning where the model  learns from the easier samples at first and proceeds to learn from difficult and challenging samples. The authors should discuss their difference with self-paced learning.

The method is positioned as a general one for feature learning. I do not know the reason why the authors only apply for object detection on a very specific dataset. It is expected to see whether the proposed method is also effective for image classification.

More datasets for evaluation are needed, even only for the object detection application.

---

> ### Author Response · Authors · 2018-11-24
> **Thanks for the comment (R2)!**
>
> Thank you for the comments. We really appreciate it and have uploaded a newer version of the manuscript (fonts marked as blue where there is a major change).
>
> >>> Authors should discuss the difference with self-paced learning.
>
> [RESPONSE]: With all due respect, we might doubt there is little resemblance of our method to the self-paced learning [kumar2010] except the fact that, both methods have two sets.
>
> [kumar2010] Kumar, M.; Packer, B.; and Koller, D. Self-paced learning for latent variable models. In NIPS 2010.
>
> (Since the reviewer doesn’t give a specific reference to self-paced learning (SPL), we looked into the literature and found the reference above that might be the most relevant for consideration)
>
> (1) SPL uses one (easy one) of the two sets first and then involves the second (hard one) to optimize the objective function. The division of sets is based on the easiness of samples and can be deemed as a way of *separating samples*, in order to optimize hidden variables. There is no *interaction* between two sets in SPL.
>
> (2) The division of sets for our method is motivated by the fact that one set is reliable and the other is less reliable; they share similar feature distribution. Thus these two sets can mutually help each other towards a more compact cluster within one class (mainly the reliable one helped the less reliable one). It is deemed as a way of interacting two sets from the very start during the training process, in order to optimize the class centroid in feature space. Our method always *interact* with the two sets simultaneously by the inherent design in feature intertwiner.
>
> (3) Besides, the definition of sets in [kumar2010] is also different than ours. The easy examples (which they define as those that generate a smaller loss, and refers to the “reliable set” in our case) could be a small object, corresponding to the less reliable set, which is contradictory in our setting by definition.
>
> Therefore, we deem these two works are parallel to each other.
>
> We have added the comparison of our work with SPL at the end of the first paragraph in related work, Sec. 6.1 in the appendix.
>
>
> >>> The method is positioned as a general one for feature learning. Why only apply for object detection? It is expected to see whether it works for classification. More datasets are needed even only for object detection.
>
> [RESPONSE]: The idea of utilising one reliable set to help the feature learning of the less reliable set is very general for feature learning. For now, we apply to object detection only because it is naturally curated for generating these two sets: based on their size/resolution compared with RoI output’s size.
>
> As for classification, it’s important to divide features from the input into one reliable set and the other less reliable one. It requires some discussions and studies, which is not our focus in this paper. Instead of investigating how to *generate* the two sets, we mainly discuss how to *use* information from the reliable set to guide the learning of the less reliable set.  We will leave the possible extension to image classification as future work.
>
> We have added one more paragraph in the last conclusion section (Sec. 5) to address reviewer’s concern on this.
>
> Please note that the title of Section 3 is already shrunk to “Feature Intertwiner for Object Detection” in the first submission. The introduction describes the idea of feature intertwiner in general while the method and experiment sections narrow down the problem into the object detection. We can change the paper’s title as “Feature Intertwiner for Object Detection” based on reviewer’s suggestion.
>
> [MORE DATASET FOR DETECTION]: We have conducted experiments on another standard object detection dataset to verify our idea. Please see Table 5 in the revised manuscript. On Pascal VOC 2007 dataset, we achieve a mAP of 73.1%, compared with SSD 71.6%, Faster-RCNN 69.9% and Fast R-CNN 66.9%.

---

### Meta-Review · Area_Chair1 · 2018-12-14
**decision**

**Confidence:** 4
**Recommendation:** Accept (Poster)

**Metareview:**

The paper proposes an interesting idea (using "reliable" samples to guide the learning of "less reliable" samples). The experimental results and detailed analysis show clear improvement in object detection, especially small objects.

On the weak side, the paper seems to focus quite heavily on the object detection problem, and how to divide the data into reliable/less-reliable samples is domain-specific (it makes sense for object detection tasks, but it's unclear how to do this for general scenarios). As the authors promise, it will make more sense to change the title to "Feature Intertwiner for Object Detection" to alleviate such criticisms.

Given this said, I think this paper is over the acceptance threshold and would be of interest to many researchers.